# Precise base editing for the *in vivo* study of developmental signaling and human pathologies in zebrafish

**Marion Rosello[1,2], Juliette Vougny[2], François Czarny[1], Marina C Mione[3], Jean-Paul Concordet[4], Shahad Albadri[1]\*, Filippo Del Bene[1,2]\***

[1]Sorbonne Université, INSERM, CNRS, Institut de la Vision, Paris, France; [2]Institut Curie, PSL Research University, Inserm U934, CNRS UMR3215, Paris, France; [3]Department of Cellular, Computational and Integrative Biology – CIBIO, University of Trento, Trento, Italy; [4]Muséum National d'Histoire Naturelle, INSERM U1154, CNRS UMR 7196, Paris, France

**Abstract** While zebrafish is emerging as a new model system to study human diseases, an efficient methodology to generate precise point mutations at high efficiency is still lacking. Here we show that base editors can generate C-to-T point mutations with high efficiencies without other unwanted on-target mutations. In addition, we established a new editor variant recognizing an NAA protospacer adjacent motif, expanding the base editing possibilities in zebrafish. Using these approaches, we first generated a base change in the *ctnnb1* gene, mimicking oncogenic an mutation of the human gene known to result in constitutive activation of endogenous Wnt signaling. Additionally, we precisely targeted several cancer-associated genes including *cbl*. With this last target, we created a new zebrafish dwarfism model. Together our findings expand the potential of zebrafish as a model system allowing new approaches for the endogenous modulation of cell signaling pathways and the generation of precise models of human genetic disease-associated mutations.

**\*For correspondence:**
shahad.albadri@inserm.fr (SA);
filippo.del-bene@inserm.fr (FDB)

**Competing interests:** The authors declare that no competing interests exist.

## Introduction

With the recent technological advances in precise gene editing, the use of zebrafish in genetic engineering studies has drastically increased in the last 5 years (**Patton and Tobin, 2019**; **Santoriello and Zon, 2012**). The CRISPR (clustered regularly interspaced short palindromic repeats)/Cas9 system is indeed a remarkably powerful gene-editing tool (**Sander and Joung, 2014**) that enables the rapid and efficient generation of loss-of-function mutations in this animal model. This system relies on the specific binding of a sgRNA-Cas9 complex that initially interacts with DNA 20 base pair (bp) upstream of a NGG protospacer adjacent motif (PAM) sequence that triggers the Cas9 protein to introduce a double-strand break (DSB). This technique is nowadays widely used in zebrafish notably to produce knock-out alleles (**Hwang et al., 2013**), and more recently, it has also been demonstrated that CRISPR/Cas9-mediated homology-directed repair (HDR) can be used to introduce exogenous DNA and single-nucleotide polymorphisms (**Prykhozhij et al., 2018**; **Tessadori et al., 2018**; **Wierson et al., 2020**).

Recently, a CRISPR/Cas9-based technology has been developed to precisely edit single bases of DNA without introducing DSBs in human cells (**Koblan et al., 2018**; **Komor et al., 2016**; **Komor et al., 2017**). The method is based on the fusion of a Cas9-D10A nickase with a cytidine deaminase giving rise to a cytidine base editor (CBE). CBE converts C-to-T bases in a restricted window of 13–19 nucleotides (nt) upstream of the PAM sequence (**Figure 1A**). In zebrafish, a CBE was shown to work but with limited efficiencies, inducing less than 29% of gene editing and, in most

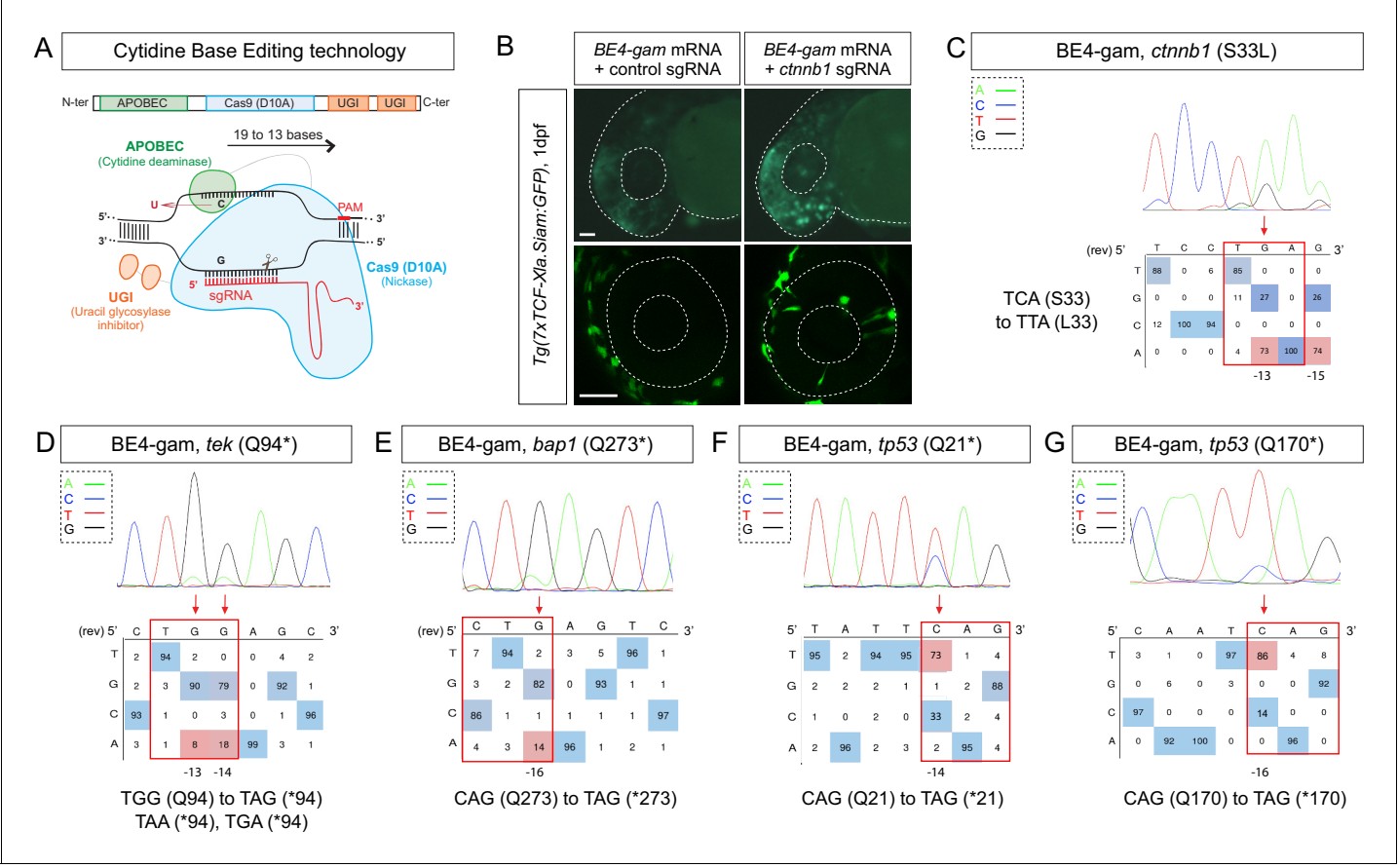

**Figure 1.** Efficient endogenous activation of Wnt signaling pathway and tumor suppressor genes targeting using BE4-gam in zebrafish. (**A**) Schematic representation of the cytidine base editor technology. (**B**) Activation of Wnt signaling via S33L mutation in β-catenin. 1 dpf *Tg(7xTCF-Xla.Siam:GFP)* representative embryos injected with *BE4-gam* mRNA and *ctnnb1* (*S33L*) sgRNA or control scrambled sequence. The upper panel shows an overall increase of GFP-positive cells in the head/anterior region upon the injection of the *BE4-gam* mRNA and *ctnnb1* (*S33L*) sgRNA compared to the control situation. The lower panel shows maximal z-projection of lateral view of the injected embryos where ectopic GFP signal in retinal progenitor cells (white stars) can be detected, whereas control embryos do not show any fluorescence in the retina at this stage. (**C–G**) DNA sequencing chromatogram of targeted loci with the BE4-gam and obtained C-to-T conversion efficiencies. The chromatograms correspond to the highest efficiency reported for the single embryos analyzed as detailed in *Table 2*. (**C**) S33L mutation in β-catenin upon C-to-T conversion in *ctnnb1* reached 73% of gene-editing efficiency. The other edited C led to a silent mutation GAC (**D**) to GAT (**D**). (**D**) Q94* mutation in Tek upon C-to-T conversion in *tek* reached 18% of gene-editing efficiency. (**E**) Q273* mutation in Bap1 upon C-to-T conversion in *bap1* reached 14% of gene-editing efficiency. (**F**) Q21* mutation in p53 upon C-to-T conversion in *tp53* reached 73% of gene-editing efficiency. (**G**) Q170* mutation in p53 upon C-to-T conversion in *tp53* reached 86% of gene-editing efficiency. For (**C**) and (**E**), the reverse complement of the sgRNA sequence is shown. Scale bars: (**B**) 50 μm. (**D–G**) Numbers in the boxes represent the percentage of each base at that sequence position. In red are highlighted the base substitutions introduced by base editing, while the original bases are in blue. The color code of the chromatogram is indicated in the upper left corner (Adenine green, Cytosine blue, Thymine red, Guanine black). The distance from the PAM sequence of the targeted C base is indicated below each chromatogram. It is considered that the quantifications under 5% are due to the background signal from Sanger sequencing and are thus non-significant (*Kluesner et al., 2018*).

The online version of this article includes the following figure supplement(s) for figure 1:

**Figure supplement 1.** List of targeted loci.

cases, at least 5% of unwanted INDEL (insertion or deletion) mutations were also detected (*Carrington et al., 2020*; *Zhang et al., 2017*). For these reasons, this editing strategy has not been so far favored by the zebrafish community. However, since this first generation of CBEs, several studies in cell culture have optimized and engineered new base editor variants with increased gene-editing efficiency reaching up to 90% without creating undesired INDEL mutations (*Koblan et al., 2018*). A recent study reported the use of a second-generation CBE to generate a zebrafish model of human ablepharon macrostomia syndrome (*Zhao et al., 2020*).

Recent progress has also been made on the generation of CBEs able to recognize other PAM sequences, allowing to broaden gene-editing possibilities (*Jakimo et al., 2018*; *Koblan et al., 2018*). Thus, base editing offers a complementary and powerful approach in zebrafish to introduce specific single-nucleotide variants into the zebrafish genome. Here based on these technological advances, we optimized these second-generation gene-editing tools in zebrafish. As reported in *ex vivo* studies (*Koblan et al., 2018*), we tested different CBE variants and obtained highly efficient C-to-T conversion, reaching up to 91% efficiency without unwanted mutations and expanded base editing possibilities using a CBE variant recognizing the NAA PAM. Furthermore, compared to previous studies, here we used these tools to target Wnt signaling, thus proving that endogenous pathways can be modulated in their natural context. Finally, we demonstrated the power of this technology for introducing precise mutations in human cancer-associated genes with high efficiency in zebrafish and created a new fish model for dwarfism.

## Results and discussion

### BE4-gam base editing for the endogenous activation of Wnt signaling pathway

To date, the main strategies used in zebrafish to study the constitutive activation of signaling pathways and to dissect their role during embryonic development or tumorigenesis were based on overexpressing mutated genes. To gain further insights and to complement these studies, an important requirement is to have the ability to maintain the endogenous genetic and regulatory contexts by generating mutations of endogenous genes *in vivo*.

To address this challenge, we decided to introduce an activating mutation in the *ctnnb1* gene coding for the key effector β-catenin of canonical Wnt signaling, a major signaling pathway during embryonic development which is activated in many cancers (*Steinhart and Angers, 2018*). It was previously shown that the mutation of the Serine33 of the human β-catenin protein into a Leucine prevents its degradation by the ubiquitin-proteasome system, leading to its stabilization and thereby to the constitutive activation of Wnt signaling pathway (*Hart et al., 1999*; *Liu et al., 1999*).

We first aimed at introducing this mutation in the genome of the zebrafish by using the Base Editor 4 fused to the gam domain (BE4-gam) (*Figure 1A*). This CBE was indeed one of the first variants of CBEs to show high efficiency of gene editing and fewer INDELs formation in cultured cells (*Komor et al., 2017*). We injected the *BE4-gam* mRNA and synthetic *ctnnb1 S33L* sgRNA into one-cell stage *Tg(7xTCF-Xla.Siam:GFP)* zebrafish embryos to directly monitor the effect of the introduced mutation on the activity of the canonical Wnt signaling (*Moro et al., 2012*). Upon *ctnnb1 S33L* sgRNA injection, we observed an increase of GFP-positive cells at 1 dpf (n = 39/50 embryos) compared to the control embryos (n = 27 embryos) resulting from three independent experiments. By confocal imaging we quantified ectopic activation of the pathway in retinal progenitor cells and observed an average of 12 GFP-positive clones per retina, while GFP-positive cells were never detected in the retina of control injected *Tg(7xTCF-Xla.Siam:GFP)* embryos (n = 4 each) (*Figure 1B*). Using this strategy, we observed base editing in five of eight randomly chosen embryos and were able to reach up to 73% of editing efficiency in single embryo analysis (*Table 1*, *Table 2*). In addition, we also observed the conversion of another cytidine within the PAM [−19, −13 bp] window leading to a silent mutation (GAC-to-GAT (D)) in four of the eight analyzed embryos with up to 74% efficiency (*Figure 1C*, *Table 1*, *Table 2*, *Figure 1—figure supplement 1*).

With these results, we demonstrated that it is now possible to constitutively and efficiently activate important developmental signaling pathways in their endogenous context, as we show here for Wnt signaling. Furthermore, several studies have implicated the S33L β-catenin mutation in tumorigenesis, making it possible to study the role of this oncogenic mutation in cancer development in zebrafish. In order to test the potential of CBE targeting in cancer modeling, we next decided to use it to target a series of tumor suppressor genes and oncogenes using the same editing strategy applied to endogenous β-catenin.

### Base-editing strategies for the generation of human cancer mutations

Zebrafish is a powerful model system to study cancer genetics *in vivo* (*Cagan et al., 2019*; *Cayuela et al., 2018*). However, a robust method for modeling cancer-associated mutations in

**Table 1.** Base-editing efficiency using different CBE variants.

Number of edited embryos randomly chosen after injection of *CBE* mRNA and sgRNA. The efficiency varies between non-detected (n. d.) and 91% depending on the targeted locus, the sgRNA, and the CBE used. Editing efficiency was quantified by editR analysis (*Kluesner et al., 2018*), which does not detect editing efficiency below 5%.

| Targeted gene / CBE used / induced mutation | ctnnb1 (S33L) BE4-gam | tp53 (Q170*) BE4-gam | cbl (W577*) BE4-gam | | kras (E62K) BE4-gam | Kras (E62K) ancBE4max | dmd (Q8*) BE4-gam | dmd (Q8*) ancBE4max | rb1 (W63*) ancBE4max | | nras (G13S) spymac-ancBE4max | tp53 (Q170*) spymac-ancBE4max |
|---|---|---|---|---|---|---|---|---|---|---|---|---|
| Number of edited embryos | 5/8 | 7/8 | 8/10 | | 0/8 | 4/7 | 0/8 | 2/4 | 8/8 | | 2/4 | 1/4 |
| Highest obtained efficiency | 73% | 86% | C16 35% | C15 50% | n.d. | 19% | n.d. | 14% | C17 91% | C16 65% | 19% | 16% |

**Table 2.** Editing efficiency quantification.

Editing quantification of up to 10 single embryos randomly chosen after injection of indicated *CBE* mRNA and sgRNA. The efficiency varies between non-detected (n.d.) to 91% in a single embryo depending on the targeted locus, the sgRNA, and the CBE used. Editing efficiency was quantified by editR analysis (*Kluesner et al., 2018*), which does not detect editing efficiency below 5%.

| Targeted gene / CBE used | Number of edited embryos | Emb. 1 | Emb. 2 | Emb. 3 | Emb. 4 | Emb. 5 | Emb. 6 | Emb. 7 | Emb. 8 | Emb. 9 | Emb. 10 |
|---|---|---|---|---|---|---|---|---|---|---|---|
| ctnnb1 (S33L) BE4-gam | 5/8 | C15 74% / C13 73% | C15 n.d. / C13 40% | C15 44% / C13 25% | C15 7% / C13 16% | C15 n.d. / C13 11% | n.d. | n.d. | n.d. | – | – |
| tek (Q94*) BE4-gam | 5/8 | C14 18% / C13 8% | C14 10% / C13 n.d. | C14 8% / C13 n.d. | C14 6% / C13 n.d. | C14 8% / C13 9% | n.d. | n.d. | n.d. | – | – |
| Bap1 (Q273*) BE4-gam | 4/8 | 14% | 12% | 9% | 8% | n.d. | n.d. | n.d. | n.d. | – | – |
| tp53 (Q21*) BE4-gam | 6/8 | 63% | 33% | 37% | 58% | 8% | 50% | n.d. | n.d. | – | – |
| tp53 (Q170*) BE4-gam | 7/8 | 86% | 46% | 51% | 62% | 45% | 20% | 33% | n.d. | – | – |
| cbl (W577*) BE4-gam | 8/10 | C16 35% / C15 50% | C16 19% / C15 31% | C16 22% / C15 38% | C16 25% / C15 41% | C16 20% / C15 35% | C16 7% / C15 9% | C16 7% / C15 12% | C16 10% / C15 17% | n.d. | n.d. |
| kras (E62K) BE4-gam | 0/8 | n.d. | n.d. | n.d. | n.d. | n.d. | n.d. | n.d. | n.d. | – | – |
| kras (E62K) ancBE4max | 4/7 | C17 19% / C16 21% | C17 8% / C16 11% | C17 6% / C16 8% | C17 9% / C16 10% | n.d. | n.d. | n.d. | – | – | – |
| dmd (Q8*) BE4-gam | 0/8 | n.d. | n.d. | n.d. | n.d. | n.d. | n.d. | n.d. | n.d. | – | – |
| dmd (Q8*) ancBE4max | 2/4 | 14% | 6% | n.d. | n.d. | – | – | – | – | – | – |
| sod2 (Q145*) ancBE4max | 8/8 | 64% | 45% | 21% | 54% | 52% | 24% | 26% | 33% | – | – |
| rb1 (W63*) ancBE4max | 8/8 | C19 n.d. / C17 91% / C16 65% | C19 21% / C17 79% / C16 75% | C19 n.d. / C17 27% / C16 18% | C19 13% / C17 81% / C16 60% | C19 8% / C17 48% / C16 33% | C19 13% / C17 76% / C16 64% | C19 13% / C17 78% / C16 69% | C19 21% / C17 77% / C16 63% | – | – |
| nras (G13S) spymac-ancBE4max | 2/4 | 19% | 18% | n.d. | n.d. | – | – | – | – | – | – |
| tp53 (Q170*) spymac-ancBE4max | 1/4 | 16% | n.d. | n.d. | n.d. | – | – | – | – | – | – |

zebrafish is still lacking to date. We decided to create predictable premature stop codons in tumor suppressor genes and to generate activating mutations in oncogenes of the RAS family (*Li et al., 2018*) in order to test the ability of CBEs to induce cancer-related mutations in zebrafish.

We first developed an automated script to rapidly detect codons allowing to generate nonsense mutations after a C-to-T conversion within the restricted PAM [−19, −13] bp editing window (*Source code 1*). Using this script, we designed a series of sgRNAs targeting in a selection of tumor suppressor genes. We induced the Q94* mutation in Tek in five of eight randomly chosen embryos, Q273* mutation in Bap1 in four of eight randomly chosen embryos and Q21* mutation in p53 in six of eight randomly chosen embryos as well as Q170* in p53 in seven of eight randomly chosen embryos by C-to-T conversions (*Figure 1D–G*, *Table 1*, *Table 2*). Among the different targeted mutations, the highest efficiency was achieved with the *tp53* tumor suppressor gene, for which we reached up to 86% of C-to-T conversion for the introduction of the Q170* mutation (*Figure 1G*, *Table 1*, *Table 2*). To assess the presence of INDELs or unwanted mutations upon BE4-gam injections in our targets, we amplified, cloned, and sequenced all targeted loci. For *tek* 6 of 20, *bap1* 2 of 12, *tp53* (Q21*) 12 of 24, and lastly *tp53* (Q170*) 21 of 24 colonies showed precise C-to-T conversions, whereas all the other analyzed sequences were wild type, without any error or INDEL formation. Together these results show that, using BE4-gam, we efficiently targeted several genes implicated in tumorigenesis in zebrafish without generating any unwanted INDELs, unlike what was previously reported with the BE3 variant.

More recently, a new CBE variant, the ancBE4max, has been engineered and optimized in cell culture with increased efficiency compared to the classical BE4-gam, reaching up to 90% efficiency and very low rates of INDELs (*Koblan et al., 2018*). We therefore decided to use this new CBE variant to target the oncogenic mutation E62K in Kras and induce the creation of a Q8* stop codon in the Dmd tumor suppressor for which we did not obtain any C-to-T conversions or other unwanted changes using the BE4-gam in randomly chosen eight embryos for each condition (*Table 1*, *Table 2*). By co-injection of *ancBE4max* mRNA with the *kras E62K* sgRNA, we were able to introduce the E62K mutation in four of seven randomly chosen embryos and we were able to reach up to 19% of editing efficiency in single embryo analysis. Another cytidine in the editing window was also converted and led to the generation of a silent mutation (CAG-to-CAA (Q)) (*Figure 2A*, *Table 1*, *Table 2*). Similar to what we observed in the case of *kras* editing, we were able to obtain a Q8* mutation in the Dmd tumor suppressor in two of four randomly chosen embryos with up to 14% of editing efficiency (*Figure 2B*, *Table 1*, *Table 2*). Thus, with this new ancBE4max variant, we are able to introduce mutations that could not be achieved with BE4-gam using the same sgRNAs. Remarkable editing efficiency was also observed using this CBE for two additional targets: the tumor suppressor genes *sod2* and *rb1*, for which, respectively, up to 64% and 91% of editing were reached and 100% of the sequenced embryos were precisely mutated (from single embryo analysis of n = 8 randomly analyzed embryos) (*Figure 2C,D*, *Table 1*, *Table 2*; *Bravard et al., 1992*; *Dyson, 2016*).

It is interesting to note that in general all the cytidine bases present in the PAM [−19, −13] bp window can be edited by the CBE, with a higher efficiency for the cytidine bases located in the middle of this window while editing was below detection levels for cytidines located only 12 bp upstream (*Figure 2D*). With the use of ancBE4max CBE, these results highlight the importance of the cytidine distance from the PAM for efficient editing in zebrafish as shown previously in cell culture assays (*Gaudelli et al., 2017*).

## Expanding gene-editing possibilities in zebrafish using a CBE recognizing NAA PAM

Due to the PAM-dependent restriction of the editing window, many mutations could not be generated so far. We therefore decided to expand the editing possibilities in zebrafish by associating *Spymac*Cas9 recognizing NAA PAMs with the efficient conversion capacity of the ancBE4max. To this end, we replaced the PAM-interacting motif (PIM) domain of the *Sp*Cas9 with the one of the *Spymac*Cas9 in the ancBE4max (*Jakimo et al., 2018*). The inserted PIM domain was codon optimized for zebrafish. Using this newly generated ancBE4max-*Spymac*Cas9, we were able to reproduce the human G13S mutation in Nras oncogene in zebrafish in 2 out of 4 randomly analyzed embryo with up to 19% of efficiency reached in single embryo analysis (*Figure 2E*, *Table 1*, *Table 2*). We also introduced a stop codon by a C-to-T conversion in the *tp53* gene in 1 out of 4 randomly analyzed embryo with 16% efficiency (*Figure 2F*, *Table 1*, *Table 2*). These results demonstrate that in addition

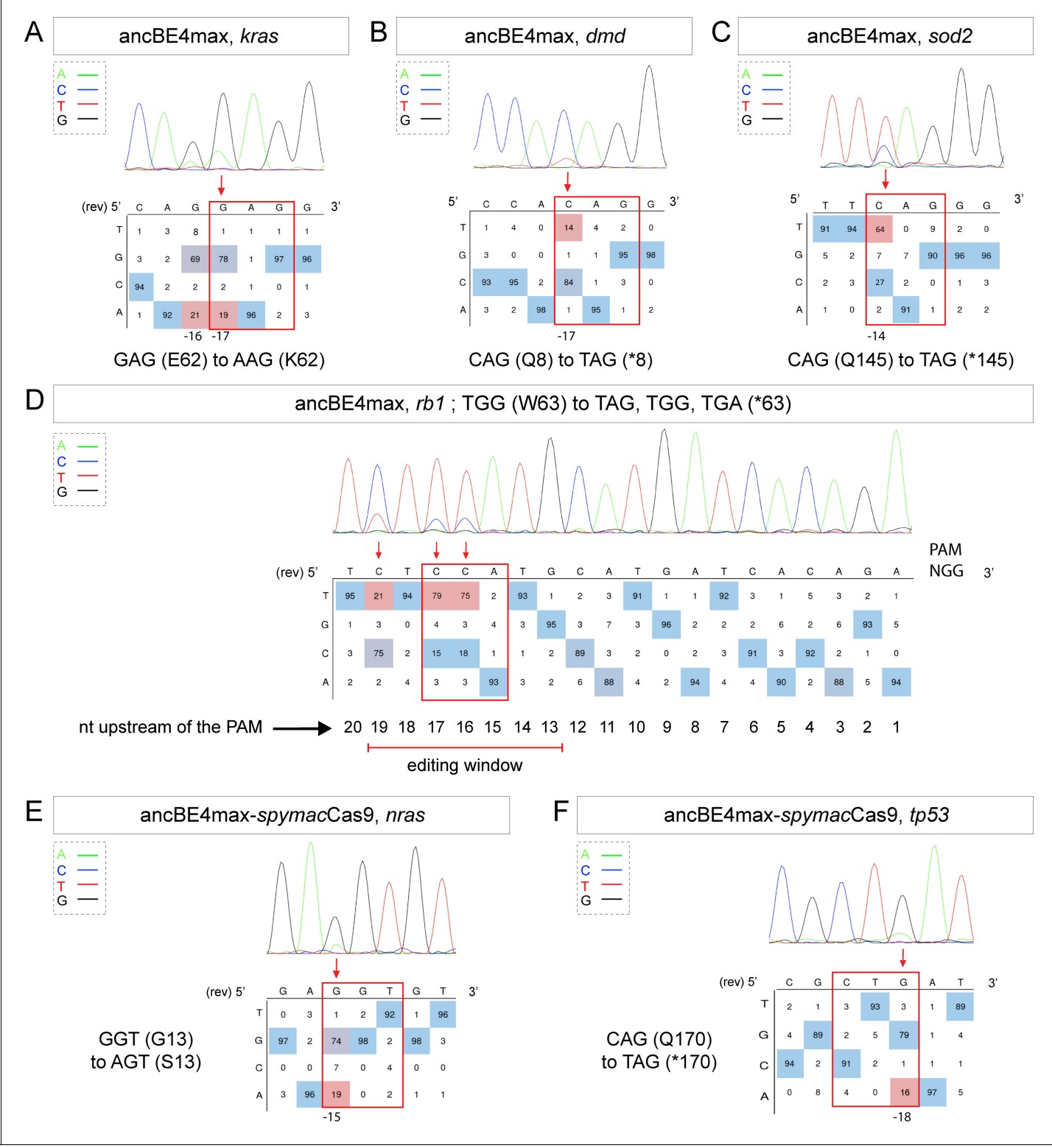

**Figure 2.** Tumor suppressor genes and oncogenes targeting by the highly efficient ancBE4max and the ancBE4max-*Spymac*Cas9 recognizing NAA PAM. (A–F) DNA sequencing chromatogram of targeted loci with the ancBE4max (in **A**–**D**) or ancBE4max-*Spymac*Cas9 (in **E**,**F**) and obtained C-to-T conversion efficiencies. (**A**) E62K mutation in Kras upon C-to-T conversion in *kras* reached 19% gene-editing efficiency. The other edited C led to a silent mutation CAG (Q) to CAA (Q). (**B**) Q8* mutation in Dmd upon C-to-T conversion in *dmd* reached 14% of gene-editing efficiency. (**C**) Q145* mutation in Sod2 upon C-to-T conversion in *sod2* reached 64% of gene-editing efficiency. (**D**) W63* mutation in Rb1 upon C-to-T conversion in *rb1*

*Figure 2 continued on next page*

Figure 2 continued

reached 21% for the C19 base, 79% for C17, and 75% for the C16 of gene-editing efficiency. (E) G13S mutation in Nras upon C-to-T conversion in *nras* reached 19% of gene-editing efficiency. (F) Q170* mutation in p53 upon C-to-T conversion in *tp53* reached 16% of gene-editing efficiency. For (A, D–F), the reverse complement of the sgRNA sequence is shown. (A–F) The chromatograms correspond to the efficiency reported for the single embryos provided in the first column of *Table 2*. The numbers in the boxes represent the percentage of each base at that sequence position. In red are highlighted the base substitutions introduced by base editing, while the original sequence is in blue. The color code of the chromatogram is indicated in the upper left corner (Adenine green, Cytosine blue, Thymine red, and Guanine black). The distance from the PAM sequence of the targeted C base is indicated below each chromatogram. It is considered that the quantifications under 5% are due to the background signal from Sanger sequencing and are thus non-significant (*Kluesner et al., 2018*).

to the classical NGG PAM it is now also possible to target NAA PAMs in zebrafish, thereby significantly expanding the range of cytidine bases that can be converted. For these new CBEs, we added a function in our script to choose the PAM recognized by the Cas9-D10A of the chosen CBE to generate the desired base editing (*Source code 1*). Together with the use of the ancBE4max and ancBE4max-*Spymac*Cas9 CBE variants, we were now able to target mutations that could not be generated with the BE4-gam base editor and reproduce a wider range of human cancer mutations in zebrafish.

Genetic alterations that lead to oncogene activation and/or tumor suppressor inactivation are responsible for tumorigenesis. It is indeed well-established that in cancer patients, a series of genetic mutations in tumor suppressor genes and/or oncogenes are combined to all together lead to the appearance of the disease (*Dash et al., 2019*). With these efficient genetic tools that are now established in zebrafish, we have the possibility to rapidly test precise combinations of mutations identified in cancer patients.

## Precise gene editing in the *cbl* tumor suppressor gene for the generation of human disease phenotypes in zebrafish

With the technological advances in CRISPR/Cas9 gene editing, zebrafish has become an even more attractive system for modeling human genetic diseases. Among the chosen loci to test the efficiency of the BE4-gam, we targeted the tumor suppressor gene encoding for Cbl, an E3 ubiquitin ligase, that is found mutated in Noonan syndrome patients presenting short stature and other bone malformations among several other phenotypes (*Martinelli et al., 2010*). In human, activating mutations in the *fibroblast growth factor receptor 3* (*FGFR3*) gene are a leading cause of dwarfism achondroplasia and related dwarf conditions. Indeed, FGFR3 hyperactivation triggers intracellular signaling within the chondrocytes of the growth plate which terminates its proliferation and bone growth (*Harada et al., 2009*). Interestingly, another study based on *in vitro* systems reported that some of these activating mutations in *FGFR3* disrupt c-Cbl-mediated ubiquitination that serves as a targeting signal for lysosomal degradation and termination of receptor signaling (*Cho et al., 2004*). Using the CBE BE4-gam as previously described, we obtained up to 50% of gene-editing efficiency (*Figure 3A*, *Table 1*, *Table 2*), with 80% of the analyzed embryos showing the expected editing (n = 10 randomly analyzed embryos). Four of 15 adults carried the Cbl W577* mutation in germ cells and one of these carriers transmitted it to 28% of its F1 offspring (44 of 153 analyzed fish carried the mutation). The target sequence was analyzed in the F1 embryos and no INDELs were found (*Figure 3B*). The zygotic homozygous mutant fish ($cbl^{-/-}$) deriving from the incross of two heterozygous parents ($cbl^{+/-}$) did not develop any obvious phenotype and could be grown to adulthood. This could be due to the fact that in zebrafish, maternal factors stored as mRNAs and proteins in the egg can compensate for zygotic loss of function during embryonic stages. In order to obtain maternal-zygotic mutants ($MZ\ cbl^{-/-}$) that lacked wild-type *cbl* mRNAs and proteins provided by the mother, $cbl^{-/-}$ mutant parents were incrossed. As controls, the $cbl^{+/+}$ siblings of the $cbl^{-/-}$ mutant fish were incrossed in parallel. Interestingly, 24% of the $MZ\ cbl^{-/-}$ mutants displayed a significantly reduced overall growth and size by 3 months post-fertilization, while 100% of the progeny of the $cbl^{+/+}$ sibling fish showed a normal body size (means: 2.7 cm for the wild-type controls and 1.96 cm for the dwarf $MZ\ cbl^{-/-}$, *Figure 3C,D*). Furthermore, this dwarf phenotype was never observed in any of the fish derived from the incrosses of the wild-type stocks used to generate this mutant line, while it was observed in the progeny of two other crosses of the $cbl^{-/-}$ line. Although we cannot formally exclude the presence of a distinct maternal zygotic mutation linked to the $cbl^{W577*}$ allele, our

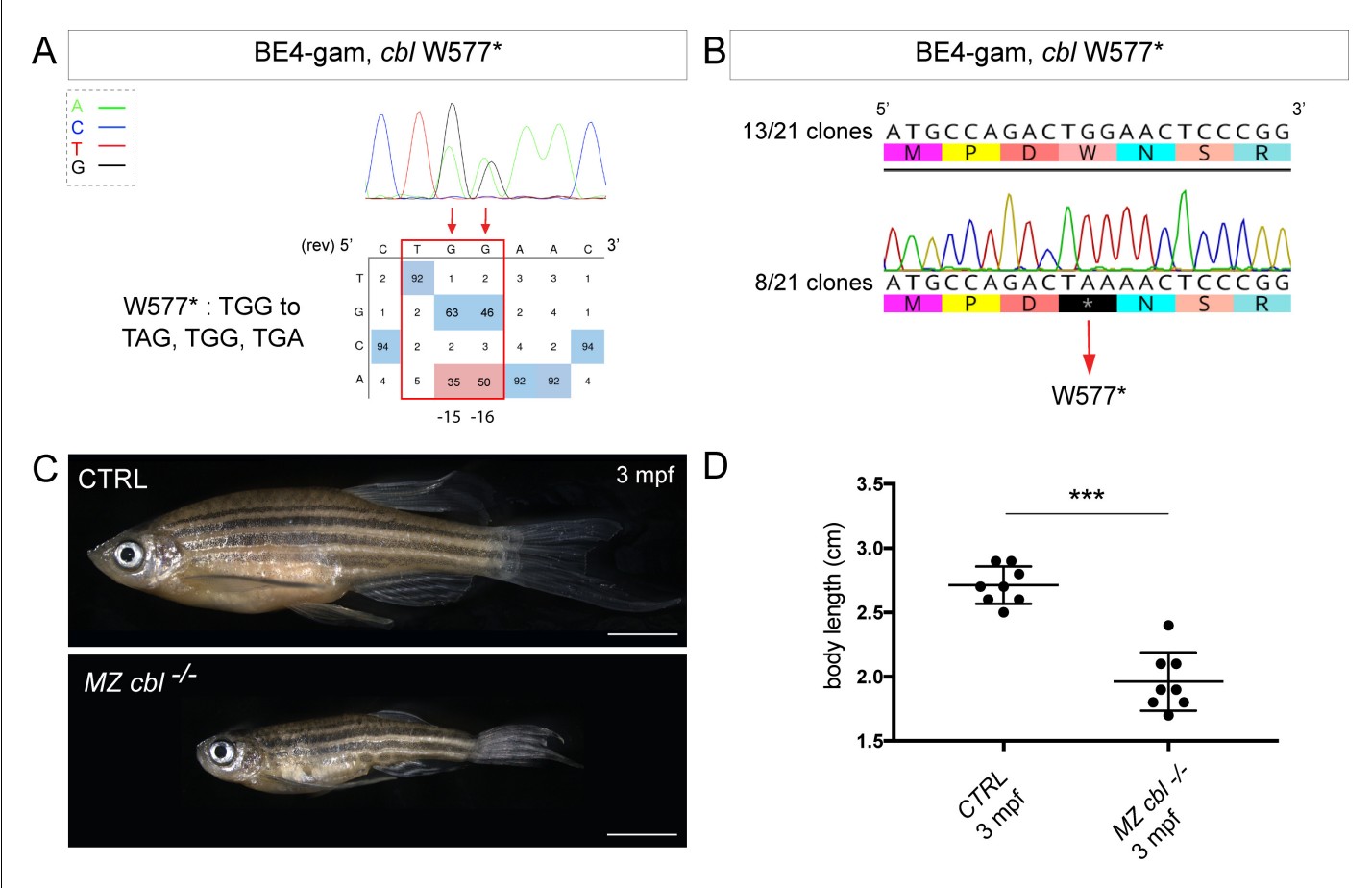

**Figure 3.** BE4-gam generated *cbl* maternal zygotic mutant fish show a reduced growth phenotype. (**A**) DNA sequencing chromatogram of targeted *cbl* gene with the BE4-gam. W577* mutation in Cbl upon C-to-T conversion in *cbl* reached 50% for the C16 base and 35% for the C15 base of gene-editing efficiency. The chromatogram refers to the efficiency reported for the embryo provided in the first column of *Table 2*. The numbers in the boxes represent the percentage of each base at that sequence position. In red are highlighted the base substitutions introduced by base editing, while the original sequence is in blue. The color code of the chromatogram is indicated in the upper left corner (Adenine green, Cytosine blue, Thymine red, and Guanine black). The distance from the PAM sequence of the targeted C base is indicated below the chromatogram. It is considered that the quantifications under 5% are due to the background signal from Sanger sequencing and are thus non-significant (*Kluesner et al., 2018*). (**B**) Sequencing of individual clones of a pool of F1 embryos from a founder carrying the W577* mutation in Cbl. TGG-to-TAA precise mutation was found in 8 of 21 clones. No editing or INDELs were detected in all other clones. (**C**) Three months post-fertilization (mpf) *cbl* wild type derived from the incross of wild-type siblings (upper panel) and dwarf maternal zygotic (MZ) mutant fish found in 24% of the progeny (lower panel). (**D**) Quantification of the body length of the *cbl*[+/+] controls and of the dwarf *MZ cbl*[−/−]. The dwarf fish show a significant reduced size at three mpf compared to the wild-type controls. n = 8 for each group. Mann–Whitney test, p=0,0002. Scale bars: (**C**) 5 mm.

data strongly support the role of Cbl W577* in the observed phenotype. Four germline mutations located in the RING domain of Cbl (Q367P, K382E, D390Y, and R420Q) have been previously identified and associated to Noonan syndrome and related phenotypes (*Martinelli et al., 2010*). Our results are in line with the growth defect phenotypes observed in these patients and directly implicate Cbl loss-of-function as a cause of bone malformations in an animal model. In addition, a point mutation in zebrafish Cbl (H382Y) has been implicated in myeloproliferative disorders. Unlike our mutant, *cbl*[H382Y] mutant fish do not survive to adulthood, suggesting that the Cbl[W577*] premature stop reported here may have different consequences on the multiple functions of Cbl (*Peng et al., 2015*). Although not lethal, it would be of interest to assess whether any hematopoietic defects are present in our *MZ cbl*[−/−] mutants or whether this phenotype is only linked to the Cbl[H328Y]

substitution found in the *LDD731* zebrafish mutant (*Peng et al., 2015*). Our model represents a powerful *in vivo* system to dissect the role of Cbl in bone morphogenesis and to explain the human phenotypes related to bone malformations.

## Conclusions

In our work, we took advantage of base editors to generate C-to-T point mutations at unprecedented high efficiencies (up to 91%) without detecting any unwanted mutations that were often problematic when using CBEs in zebrafish. In comparison, previous work has reported an efficiency reaching a maximum of 29% using the BE3 (*Zhang et al., 2017*). Another more recent study employed the ancBE4max variant in zebrafish with a slight difference of efficiency that might be due to the choice of the specific locus targeted, the synthesis of the sgRNA (homemade *vs* commercially synthetized) and the injection mode (yolk *vs* cell) (*Carrington et al., 2020*). More recently, *Zhao et al., 2020* have shown similar efficiencies as we obtained in our study. To expand the gene-editing possibilities in this animal model, we established in addition a new editor variant recognizing the NAA PAM. Using these approaches, we first performed the endogenous and constitutive activation of Wnt signaling by introducing the S33L mutation in β-catenin. In addition, we demonstrated using these strategies that we were able to precisely target several cancer-associated genes for which so far only transgenic over-expressions or imprecise deletions were used to elucidate their functions. Among our targets, the introduced mutation in the *cbl* gene allowed us to generate a new zebrafish model for dwarfism.

Together our work provides a panel of examples whereby, using gene-editing approaches, some of which we established here in zebrafish for the first time, it is now possible to manipulate endogenous signaling pathways, to generate models for human genetic disorders and to mimic precise cancer-associated mutations in zebrafish. While a recent study reported the use of ancBE4max in zebrafish (*Zhao et al., 2020*), in our work we provide a direct comparison of BE4-gam, ancBE4max, and *Spymac*-ancBE4max. Our study highlights the power and the need for these approaches to increase the efficiency and the targeting flexibility in order to model pathological human mutations in zebrafish.

Finally, the high efficiencies of CBEs obtained in this study should encourage future applications where they could be implemented with mosaic mutation induction technologies such as the MAZER-ATI (Modeling Approach in Zebrafish for Rapid Tumor Initiation) system (*Ablain et al., 2018*). This will allow to rapidly model and study *in vivo* combinations of endogenous mutations occurring in specific cancer patients or in genetic disorders caused by somatic mosaicism. Our approach could thus be applied in zebrafish for the precise modeling of complex combinations of cancer-causing mutations in adult animal models as currently possible by transgenic overexpression or somatic gene inactivation (*Callahan et al., 2018*).

# Materials and methods

**Key resources table**

| Reagent type (species) or resource | Designation | Source or reference | Identifiers | Additional information |
|---|---|---|---|---|
| Genetic reagent (*Danio rerio*) | *Tg(7xTCF-Xla.Siam:GFP)* | ZIRC | ZFIN ID: ZBD-ALT-110113–1 | |
| Recombinant DNA reagent | pCMV_BE4-gam (plasmid) | Addgene | Addgene:#100806 RRID:Addgene_100806 | |
| Recombinant DNA reagent | pCMV_ancBE4max (plasmid) | Addgene | Addgene:#112094 RRID:Addgene_112094 | |
| Recombinant DNA reagent | pCS2+_ancBE4max-SpymacCas9 (plasmid) | This paper | | See Materials and methods |
| Commercial assay or kit | NEBuilder HiFi DNA Assembly Cloning Kit | New England Biolabs | Catalog# E5520S | |

*Continued on next page*

*Continued*

| Reagent type (species) or resource | Designation | Source or reference | Identifiers | Additional information |
|---|---|---|---|---|
| Commercial assay or kit | mMESSAGE mMACHINE T7 Ultra kit | Ambion | Catalog# AM1345 | |
| Commercial assay or kit | mMESSAGE mMACHINE Sp6 kit | Ambion | Catalog# AM1340 | |
| Commercial assay or kit | PCR clean-up gel extraction kit | Macherey-Nagel | Catalog# 740609.50 | |
| Peptide, recombinant protein | Phusion high-fidelity DNA polymerase | ThermoFisher | Catalog# F-530XL | |
| Software, algorithm | SequenceParser.py | This paper | | See *Source code 1* |

## Fish lines and husbandry

Zebrafish (*Danio rerio*) were maintained at 28°C on a 14 hr light/10 hr dark cycle. Fish were housed in the animal facility of our laboratory which was built according to the respective local animal welfare standards. All animal procedures were performed in accordance with French and European Union animal welfare guidelines. Animal handling and experimental procedures were approved by the Committee on ethics of animal experimentation. The *Tg(7xTCF-Xla.Siam:GFP)* line was kindly provided by Sophie Vriz (*Moro et al., 2012*).

## Molecular cloning

To generate the p*CS2+_ancBE4max-SpymacCas9* plasmid, the *Spymac*Cas9 PIM domain sequence has been codon optimized for expression in zebrafish using online software from IDT and synthesized with the first UGI sequence as G-block from IDT. Then, three fragments have been inserted into p*CS2+ plasmid* linearized with Xho1 using the NEBuilder HiFi DNA Assembly Cloning Kit (New England Biolabs # E5520S): a first fragment of 4161 bp of the ancBE4max to the PIM domain (amplified using the primers F-5′-CGATTCGAATTCAAGGCCTCATGAAACGGACAGCCGAC-3′ and R-5′-CGGTCTGGATCTCGGTCTTTTTCACGATATTC-3′), the Gblock fragment of 803 bp (amplified using the primers F-5′-AAAGACCGAGATCCAGACCGTGGGACAG-3′ and R-5′-TCCCGCCGCTATCCTCGCCGATCTTGGAC-3′), and a third fragment of 654 bp of the rest of the ancBE4max from the PIM domain (amplified using the primers F-5′-CGGCGAGGATAGCGGCGGGAGCGGCGGG-3′ and R-5′-CTCACTATAGTTCTAGAGGCTTAGACTTTCCTCTTCTTCTTGGGCTCGAATTCGCTGCCGTCG-3′). p*CMV_BE4-gam* (a gift from David Liu, Addgene plasmid # 100806; *Anzalone et al., 2019*) has been used to generate *BE4-gam* mRNA*in vitro*. This plasmid was linearized with Pme1 restriction enzyme and mRNAs were synthesized by *in vitro* transcription with 1 µl of GTP from the kit added to the mix, followed by Poly(A) tailing procedure and lithium chloride precipitation (using the mMESSAGE mMACHINE T7 Ultra kit #AM1345, Ambion). p*CMV_ancBE4max* (p*CMV_AncBE4max* was a gift from David Liu [Addgene plasmid # 112094]) has been linearized using AvrII restriction enzyme; mRNAs were synthesized by *in vitro* transcription with 1 µl of GTP from the kit added to the mix and lithium chloride precipitation (using the mMESSAGE mMACHINE T7 Ultra kit #AM1345, Ambion). The p*CS2+_ancBE4max-SpymacCas9* has been linearized using KpnI restriction enzyme; mRNAs were synthetized by *in vitro* transcription with 1 µl of GTP added to the mix and lithium chloride precipitation (using the mMESSAGE mMACHINE Sp6 kit #AM1340, Ambion).

## sgRNA design

A sequenceParser.py python script was developed and used to design sgRNAs for the creation of a stop codon. The first function of the script is to ask which PAM will be used to then execute the rapid detection of codons that are in the right editing windows from this predefined PAM to generate a STOP in frame after C-to-T conversion. The ORF sequence file extension is .txt and the letters in lower cases. The script can be executed from the command line interface (such as the terminal or PowerShell console) using Python version 3.

Efficiencies of sgRNAs were validated using CRISPOR online tool (*Haeussler et al., 2016*). All the synthetic sgRNAs were synthesized by IDT as Alt-R CRISPR-Cas9 crRNA and Alt-R CRISPR-Cas9 tracrRNA.

List of the crRNAs used in this study and the targeted C bases for each targeted locus. Sequences are oriented from 5′ to 3′:

| | crRNA sequence used for base editing (5′—3') |
|---|---|
| *ctnnb1* (S33L) | CTGGACTCAGGAATACACTC |
| *tek* (Q94*) | GGAGCTCCAGGTGACGGTAG |
| *bap1* (Q273*) | GACTCAGCAAGAATCAGGCC |
| *tp53* (Q21*) | AGTATTCAGCCCCCAGGTGG |
| *tp53* (Q170*) | CAATCAGCGAGCAAATTACA |
| *kras* (E62K) | CCTCCTGACCTGCAGTGTCC |
| *dmd* (Q8*) | CCACAGGACCAATGGGAGGA |
| *sod2* (Q145*) | GCTGTTCAGGGCTCAGGCTG |
| *rb1* (W63*) | TCTCCATGCATGATCACAGA |
| *nras NAA* (G13S) | AACACCTCCTGCTCCCACAA |
| *tp53 NAA* (Q170*) | ATCAGCGAGCAAATTACAGG |
| *cbl* (W577*) | AGTTCCAGTCTGGCATGTTG |

## Micro-injection

Prior injections, a mix of 2 µL of the Alt-R CRISPR-Cas9 crRNA (100 pmol/µL) and 2 µL of Alt-R CRISPR-Cas9 tracrRNA (100 pmol/µL) from IDT was incubated at 95°C for 5 min, cooled down at room temperature, and then kept on ice to form the synthetic sgRNA complex. One nanoliter of another mix containing CBE mRNA (600 ng/µL) and the synthetic sgRNA complex (43 pmol/µL) was then injected into the cell at one-cell stage zebrafish embryos.

## Genotyping

To genotype the *cbl* mutant line, a PCR was performed with primers Fwd-5′-GTACGCCTGGA-GACCCATCTC-3′ and Rev-5′-CTTTTGGACTGTCATAATCCGATGC-3′. The PCR product was digested with the restriction enzyme BsrI, which cut only on the WT allele. The WT allele resulted in two fragments (300 bp and 69 bp) and the mutant allele only one fragment (369 bp).

## Whole-embryo DNA sequencing

A series between 4 and 10 single embryos randomly chosen was analyzed for each target sequence, and the embryo with the highest efficiency is shown. Generally, between 25% and 100% embryos were positive for gene editing, that is showed >16% expected sequence modification. For genomic DNA extraction, each single embryo was digested for 1 hr at 55°C in 0.5 mL lysis buffer (10 mM Tris, pH 8.0, 10 mM NaCl, 10 mM EDTA, and 2% SDS) with proteinase K (0.17 mg/mL, Roche Diagnostics) and inactivated 10 min at 95°C. To sequence and check for frequency of mutations, each target genomic locus was PCR-amplified using Phusion High-Fidelity DNA polymerase (ThermoFisher Scientific, # F-530XL). PCR products have been extracted from an agarose gel and purified (using the PCR clean-up gel extraction kit #740609.50, Macherey-Nagel), and Sanger sequencing was performed by Eurofins. Sequence analyses were achieved using ApE software and quantifications of the mutation rate done using editR online software (*Kluesner et al., 2018*). For the verification of *cbl* mutant F1 embryos, *tek*, *bap1* and *tp53* mutations, PCR fragments were subsequently cloned into the pCR-bluntII-TOPO vector (Invitrogen). Plasmid DNA was isolated from single colonies and sent for sequencing. Mutant alleles were identified by comparison with the wild-type sequence using ApE and Geneious softwares.

Primer sequences used to amplify the targeted loci:

| *tek*<br>(Q94*) | **ATCTCAGACGTGACTCTGGTGAAC** | **TTCCTGTAGCATCTTGTAGGTGTAG** |
|---|---|---|
| *bap1*<br>(Q273*) | TTGTTTATTTTTCAGGACCATGGGG | CACCTGAAGGTATTGGTGTTTCTTG |
| *tp53*<br>(Q21*) | CTTTGCATAAGAAACAACATCCCGA | GTTCAACACTGAAAACCAAAAGAGG |
| *tp53*<br>(Q170*) | ATATCTTGTCTGTTTTCTCCCTGCT | GTCCTACAAAAAGGCTGTGACATAC |
| *kras*<br>(E62K) | CGTTCCACTATGTCCACACATTTAG | AACAGTACATTTTCTGCATACTCGC |
| *dmd*<br>(Q8*) | AGGGCTCCTTCCTTTTTCTGTTTAT | TGATCGAGTTTTGATGATTTCTCCG |
| *sod2*<br>(Q145*) | GCATATGGCTGGAAATGATGAACC | GCACTTTATGTGCATTCACTGAGG |
| *rb1*<br>(W63*) | TCTGTCAACTGTTGTTTTTCCAGAC | TTCAATATCTGCCACACATACCTCA |
| *nras*<br>(G13S) | CCTTTTCTCTCTTTTTGTCTGGGTG | CGCAATCTCACGTTAATTGTAGTGT |
| *cbl*<br>(W577*) | GTACGCCTGGAGACCCATCTC | CTTTTGGACTGTCATAATCCGATGC |

## Imaging

Embryos were oriented in low-melting agarose 0.6% with an anesthetic (Tricaine 0.013%) diluted in egg solution. The inverted laser scanning confocal microscope Zeiss CLSM-LSM780 was used for high-resolution microscopy, employing a 40× water immersion objective. Z-stacks were acquired every 1–2 µm. Leica MZ10F was used to image the whole embryos the *cbl* mutant adult fish. Image analyses were performed with ImageJ software.

## Body size quantifications

Eight control wild-type siblings and eight dwarf *MZcbl*$^{-/-}$ in total were used to measure the body size using a millimetric ruler. The length measured was from mouth to trunk. A non-parametric t-test with the Mann–Whitney correction was applied to determine significance in growth. The software used was Prism 7 (GraphPad).

# Acknowledgements

We thank Sophie Vriz for sharing the *Tg(7xTCF-Xla.Siam:GFP)* transgenic line and the members of the fish-facility in Institut Curie. We also thank Céline Revenu and Viviana Anelli for early contribution. MR was supported by the Fondation pour la Recherche Médicale (FRM grant number ECO20170637481) and la Ligue Nationale Contre le Cancer. Work in the Del Bene laboratory was supported by ANR-18-CE16 'iReelAx', UNADEV in partnership with ITMO NNP/AVIESAN (National Alliance for Life Sciences and Health) in the framework of research on vision and IHU FOReSIGHT [ANR-18-IAHU-0001] supported by French state funds managed by the Agence Nationale de la Recherche within the Investissements d'Avenir program. MCM was supported by World Wide Cancer Research, grant no. 0624, and by LILT –Trento, Program five per mille (year 2014).

# Additional information

## Funding

| Funder | Grant reference number | Author |
|---|---|---|
| Agence Nationale de la Recherche | ANR-18-CE16 "iReelAx" | Filippo Del Bene |
| Agence Nationale de la Recherche | [ANR-18-IAHU-0001 | Filippo Del Bene |

| Fondation pour la Recherche Médicale | ECO20170637481 | Marion Rosello |
| --- | --- | --- |
| Ligue Contre le Cancer | | Marion Rosello |
| UNADEV/AVIESAN | | Filippo Del Bene |
| Worldwide Cancer Research | grant no. 0624 | Marina C Mione |
| LILT -Trento | Program 5 per mille | Marina C Mione |

The funders had no role in study design, data collection and interpretation, or the decision to submit the work for publication.

## Author contributions

Marion Rosello, Conceptualization, Formal analysis, Validation, Investigation, Visualization, Methodology, Writing - original draft, Writing - review and editing; Juliette Vougny, Validation, Investigation, Methodology; François Czarny, Software; Marina C Mione, Conceptualization, Writing - review and editing; Jean-Paul Concordet, Conceptualization, Supervision, Methodology, Writing - original draft, Writing - review and editing; Shahad Albadri, Conceptualization, Formal analysis, Validation, Visualization, Writing - original draft, Writing - review and editing; Filippo Del Bene, Conceptualization, Supervision, Funding acquisition, Methodology, Writing - original draft, Writing - review and editing

## Author ORCIDs

Marion Rosello (iD) https://orcid.org/0000-0003-3935-6971
Juliette Vougny (iD) http://orcid.org/0000-0002-7361-8405
Marina C Mione (iD) http://orcid.org/0000-0002-9040-3705
Shahad Albadri (iD) https://orcid.org/0000-0002-3243-7018
Filippo Del Bene (iD) https://orcid.org/0000-0001-8551-2846

## Ethics

Animal experimentation: All procedures were performed on zebrafish embryos in accordance with the European Communities Council Directive (2010/63/EU) and French law (87/848) and approved by the Sorbonne Université ethic committee (Charles Darwin) and the French Ministry for research (APAFIS agreement #21323 2019062416186982) and by the Institut Curie ethic committee and the French Ministry for research (APAFIS agreement #6031 2016070822342309).

## Decision letter and Author response

Decision letter https://doi.org/10.7554/eLife.65552.sa1
Author response https://doi.org/10.7554/eLife.65552.sa2

## Additional files

### Supplementary files

• Source code 1. SequenceParser.py STOP codon design source code. This python code highlights in capital the codons that can converted as STOP codon by C-to-T conversion with the chosen PAM sequence at the correct distance (PAM [−19, −13] bp window).

• Transparent reporting form

### Data availability

All data generated or analysed during this study are included in the manuscript and supporting files.

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
