## [Decision Letter]

**Acceptance summary:**

The manuscript by Rosello et al. represents a major advance in implementation of cutting-edge genome editing methodologies in the zebrafish. The study seeks to describe optimized tools for precise base editing in zebrafish and to demonstrate their effective application. Overall, this study demonstrates that cytosine base editing is an efficient and powerful method for introducing precise in vivo edits into the zebrafish genome, and will be of interest to scientists who use zebrafish and other genetic systems to model human development and disease.

**Decision letter after peer review:**

Thank you for submitting your article "Precise base editing for the in vivo study of developmental signaling and human pathologies in zebrafish" for consideration by *eLife*. Your article has been reviewed by three peer reviewers, and the evaluation has been overseen by a Reviewing Editor and Richard White as the Senior Editor. The following individuals involved in review of your submission have agreed to reveal their identity: Maura McGrail (Reviewer #1); Darius Balciunas (Reviewer #2).

Essential Revisions:

1) As you will see , all three reviewers are very supportive of the work. However, all have concerns about the clarity of presentation and discussion of the data, and offer some suggestions for how these aspects could be improved. Please see the individual reviews for details. These concerns should be straightforward to address.

2) Reviewer 1 suggests a number of studies in the literature that should be cited – please cite these studies or provide a response.

3) Please provide the additional details for quantitation and details of the protocols requested by reviewer 1.

4) There are some concerns over the dwarfism phenotype in the *MZ cbl ^-/-^* mutants. Please provide further genetic analysis to eliminate the possibility that this is a background mutation, or consider this possibility in your discussion. One suggestion from the reviewers is to cross a -/- female with a +/- male and see if the phenotype correlates with the genotype.

Reviewer #1 (Recommendations for the authors):

Abstract and Introduction

1) The Abstract opens with an emphasis on optimizing base editing in zebrafish, and then the application of this technology to examine signaling/cancer pathways and disease variants in human development and disease pathology. However, the results first demonstrate modeling wnt and cancer gene pathways with BE4 base editing, followed by testing an improved BE4 base editor ancBE4max, then on expanding base editor gene editing capability by altering PAM recognition, and ending with modeling human disease with BE4 base editing. The manuscript might flow better if all the BE4 data was presented, then the BE4-ancBE4max comparison, then the ancBE4maxSpmac. Possibly the authors had tried organizing the paper that way, but it didn't work well.

2) The Introduction should be edited to include citations of the latest advances in targeted integration and SNP introduction in zebrafish:

The recent publication by Zhao et al., An optimized base editor with efficient C-to-T base editing in zebrafish, December 2020 BMC Biology, should be cited in the Introduction. They reported a zAncBE4max cytosine base editor codon optimized for zebrafish, and showed it was more efficient than BE3 at precision editing at 6 loci. Zhao et al., also created a zebrafish model of human ablepharon macrostomia syndrome (AMS) using base editing to introduce the twist2 p.E78K variant. The authors could emphasize what is distinct about the current study, which is a direct comparison of BE4, ancBE4max, and ancBE4maxSpymac, in addition to the signaling pathway and disease modeling.

Please remove or edit the statement that targeted integration by HDR is not efficient in zebrafish. This is no longer the case. Efficient strategies for introduction of exogenous DNA and SNPs by HDR in the zebrafish germline has been rigorously demonstrated in the literature and should be cited: Efficient precision targeted integration of exogenous DNA directed by short homology was reported by Essner, McGrail and colleagues (Wierson et al., 2020); introduction of human tp53 SNP variants using oligonucleotide templates was reported by Berman and colleagues (Prykhozhij et al., 2018). The authors could suggest that base editing offers a complementary and powerful approach to introduce specific single nucleotide variants into the zebrafish genome, if alternative methods such as using oligonucleotide templates aren't effective at a particular locus.

Results and Discussion

1) Provide the complete spelling of BE4-gam.

2) Figure 1 The authors demonstrate highly efficient base editing with BE4-gam in 4 zebrafish cancer genes, with introduction of predictable, known cancer variants into each. It would be helpful to include the sequence of the entire target site with PAM for each gene in Figure 1.

Figure 1 B-C: The authors introduce the wnt pathway activating mutation TCA (Serine) to TTA (Leucine) into ctnnb1 B-catenin using cytidine base editor BE4-gam. To demonstrate activation of wnt signaling ctnnb1 base editing was done in the wnt pathway reporter line Tg(tcf:GFP), and the results indicate an increase in GFP signal in the developing retina at 1dpf. Figure 1B upper panel: The low contrast makes it difficult to see the images – they appear to be low magnification of the head/anterior region of the embryo. Quantification of the amount of GFP positive cells in control vs. ctnnb1 edited embryos or retina should be included, with the appropriate number of biological replicates noted.

Figure 1C shows EditR analysis of the frequency of base editing; no details on the number of embryos analyzed was included in the results although the information is in the methods (10 embryos analyzed). Since this is a techniques paper, it would be helpful to include details on how base editing was analyzed: the number of embryos, if they were analyzed singly or pooled, how the PCR amplicons were sequenced to generate sequence data for EditR analysis. The frequencies reported in Table 1 are a little misleading, since as the authors note EditR drops out samples below 5% editing efficiency. For example, 5/8 = 63%, but in reality 5/10 = 50%.

Figure 1 D-G The authors introduced nonsense C-T* mutations into three tumor suppressor genes with up to 86% efficiency. The sentence beginning "This last result is …" should be removed and replaced with an acknowledgement the similar efficiency reported by Zhao et al., 2020. Amplicons were cloned and sequenced, and the data show either precise C-T conversion or wildtype sequence without indels. But the data in the panels D-G indicates other changes at some frequency; some clarification of this difference would be helpful.

3) Figure 2. The BE4-gam base editor wasn't effective at introducing the oncogenic E62K mutation into Kras, or a Q8* nonsense mutation into the dmd tumor suppressor, so the authors tested the newer ancBE4max. It wasn't stated how it didn't work with BE4-gam, if it was a complete failure, or just indels were induced. Using the same sgRNAs ancBE4max was effective at introducing the edits at high efficiency. This is useful knowledge, if one base editor isn't working, the other could.

One of the most impactful and novel aspects of this study is the modification of ancBE4max to recognize a different PAM. The authors switched out the SpCas9 PIM domain (PAM-interacting motif) with Spymac PIM, which recognizes the PAM NAA. The data show relatively low efficiency of base editing at nras and tp53 (19% and 16%). Future work could optimize this base editor to increase efficiency.

4) Figure 3. Here the authors demonstrate efficient BE4-gam base editing in both somatic and germline targeting of the W577* mutation in the E3 ubiquitin ligase cbl. 4/15 adults carried the Cbl W577* in germ cells and transmitted it to 28% of their F1 offspring (44/153). Remarkably, all of the F1 offspring that inherited the precise W577* allele. Data from multiple loci is needed to obtain a frequency of germline transmission of precise base edited alleles.

24% of *MZ cbl ^-/-^* adults show a short stature phenotype in comparison to MZ clb ^+/+^ siblings, and the authors claim this is a new model of dwarfism. Without details on the genetics used to establish the cbl mutant line, it's unclear if the phenotype is linked to the base edited allele or is in the background. If the fish were inbred without first establishing outcrossed F2 generation adults, it's possible this is due to background mutations. More rigorous genetic analysis or inclusion of details would strengthen the validity of the disease model.

Reviewer #2 (Recommendations for the authors):

My most significant concern has to do with the way editing data is described throughout the manuscript. To take a specific example, one might infer from Figure 1C (including figure legend) that substitution of Serine 33 for Leucine in b-catenin is >70% efficient. But is it? Based on Supplementary Table 3, editing of C13 was only detected in 5/8 embryos analyzed (62.5%), with efficiencies of 73%, 40%, 25%, 16% and 11%. Thus, average editing efficiency in positive embryos is 33%. If one were to include negative embryos, editing efficiency would be ~19% – which is still very good but quite different from 70%. I believe editing efficiency ranges must be presented in the main figures, perhaps as dot charts.

Along the same lines, it is not clear how the embryo imaged in Figure 1B was selected. How many embryos were scored for ectopic activation of wnt signaling reporter and how many were found positive? To what extent (for example, what were the numbers of GFP-positive RPCs in each)? Does the photographed embryo represent "best case" or "average"?

Given that C to T base editors may mutate any Cs within the editing window, it is important to present data from Sanger sequencing of individual amplicons, similarly to data presented for tek Q94*, bap1 Q273*, tp53 Q21*, tp53 Q170* (Figure 1D-G) and cbl W577* (Figure 3B). What about the ctnnb1 S33L amplicon from Figure 1C? Do the desired change (73% efficiency) and the silent bystander substitution (74% efficiency) co-occur at random? If that were the case, one would expect ~50% of sequences with both changes and ~20% of sequences with either single change. Or do both changes co-occur at much higher than random probability, as Figure 3C suggests? This would be extremely important if the second change was not silent but introduced a missense or nonsense mutation. For those scenarios, perhaps embryo 2 and embryo 5 (Supplementary Table 3) are worth highlighting as C15 substitution is not observed in them? This applies to other edits where significant fraction of additional C->T substitutions is observed: kras (Figure 2A) and rb1 (Figure 2D).

Reviewer #3 (Recommendations for the authors):

I have only relatively minor suggestions to improve the clarity of presented data.

Figure 1B: The number of embryos with increased GFP/ total examined should be indicated in the legend.

Figure 1C-G. It took this reviewer a long time to understand how these figures worked. Once I did, I felt it was a clear way of showing the data. However, some additional description in the legend would have been very helpful for orientation. The ACTG in the top corner was initially mistaken for sequence, rather than a color code, which likely contributed significantly to my early confusion. It should also be made clearer that the numbers represent % at that site, and the box color coding system should be clearly stated.

Figure C-G. It is not clear exactly what this data is from. Is it one embryo, a pool of embryos? How many sequences analyzed? A general strategy for sequencing is discussed in the methods, but it would be helpful to know the specifics of the data presented in this figure for each mutant.

It is not clear whether the python script used for sgRNA design will be made available.

In Figure D, I am unclear how one can have wildtype +/+ siblings in a cross with maternal zygotic -/- mutants. Wouldn't the mother have to have been -/-?

---

## [Author Response]

Essential Revisions:1) As you will see , all three reviewers are very supportive of the work. However, all have concerns about the clarity of presentation and discussion of the data, and offer some suggestions for how these aspects could be improved. Please see the individual reviews for details. These concerns should be straightforward to address.

We thank the reviewers for their positive remarks and constructive comments that helped to improve our manuscript. We have addressed their concerns and have detailed our responses below.

2) Reviewer 1 suggests a number of studies in the literature that should be cited – please cite these studies or provide a response.

We added all the citations suggested by the reviewers 1 and 2.

3) Please provide the additional details for quantitation and details of the protocols requested by reviewer 1.

We have added the required quantifications and protocol details wherever needed.

4) There are some concerns over the dwarfism phenotype in the MZ cbl ^-/-^ mutants. Please provide further genetic analysis to eliminate the possibility that this is a background mutation, or consider this possibility in your discussion. One suggestion from the reviewers is to cross a -/- female with a +/- male and see if the phenotype correlates with the genotype.

The possibility that the dwarf phenotype comes from a background mutation was addressed through the generation of control fish that we obtained by keeping the wild-type siblings from the *cbl^+/-^* incross. The *cbl^+/+^* controls in Figure 3 are from the cross of these wild-type siblings, following the same procedure as for the *MZcbl^-/-^.* We never observed dwarf phenotypes in these *cbl^+/+^* fish neither in the *cbl^+/-^* and *cbl^-/-^* fish. The dwarf phenotype was also only observed in *MZ cbl^-/-^* embryos derived from two independent crosses. It should be noted that we have now kept this *cbl^+/-^* line for 5 generations in our fishroom in the same genetic background and the dwarf phenotype was only observed when *MZ cbl^-/-^* embryos were raised. In addition, the dwarf phenotype was neither observed in our wildtype stocks from which the *cbl ^-/+^* line was derived. If a background mutation happened to be linked to our mutant allele, since the dwarf phenotype was only observed in *MZ cbl^-/-^* and not in zygotic *cbl^-/-^,* the background mutation would be in a second independent maternal zygotic mutant gene which we cannot formally exclude but seems unlikely. We now discussed these data in our manuscript as suggested.

Reviewer #1 (Recommendations for the authors):Abstract and Introduction1) The Abstract opens with an emphasis on optimizing base editing in zebrafish, and then the application of this technology to examine signaling/cancer pathways and disease variants in human development and disease pathology. However, the results first demonstrate modeling wnt and cancer gene pathways with BE4 base editing, followed by testing an improved BE4 base editor ancBE4max, then on expanding base editor gene editing capability by altering PAM recognition, and ending with modeling human disease with BE4 base editing. The manuscript might flow better if all the BE4 data was presented, then the BE4-ancBE4max comparison, then the ancBE4maxSpmac. Possibly the authors had tried organizing the paper that way, but it didn't work well.

As the reviewer pointed out, we initially to organize the manuscript presenting the data following the technological advancement rather than the biological applications but this did not flow too smoothly in our opinion. We prefer to keep the organization as it is, finishing by presenting the human disease application for which we also have a stable line.

2) The Introduction should be edited to include citations of the latest advances in targeted integration and SNP introduction in zebrafish:The recent publication by Zhao et al., An optimized base editor with efficient C-to-T base editing in zebrafish, December 2020 BMC Biology, should be cited in the Introduction. They reported a zAncBE4max cytosine base editor codon optimized for zebrafish, and showed it was more efficient than BE3 at precision editing at 6 loci. Zhao et al., also created a zebrafish model of human ablepharon macrostomia syndrome (AMS) using base editing to introduce the twist2 p.E78K variant. The authors could emphasize what is distinct about the current study, which is a direct comparison of BE4, ancBE4max, and ancBE4maxSpymac, in addition to the signaling pathway and disease modeling.

According to the reviewer 1 comment, we now incorporated the findings of Zhao et al. in the Introduction and conclusion where we distinctly emphasized our novel results with respect to previous works.

Please remove or edit the statement that targeted integration by HDR is not efficient in zebrafish. This is no longer the case. Efficient strategies for introduction of exogenous DNA and SNPs by HDR in the zebrafish germline has been rigorously demonstrated in the literature and should be cited: Efficient precision targeted integration of exogenous DNA directed by short homology was reported by Essner, McGrail and colleagues (Wierson et al., 2020); introduction of human tp53 SNP variants using oligonucleotide templates was reported by Berman and colleagues (Prykhozhij et al., 2018). The authors could suggest that base editing offers a complementary and powerful approach to introduce specific single nucleotide variants into the zebrafish genome, if alternative methods such as using oligonucleotide templates aren't effective at a particular locus.

We apologize for this and have made the required modifications.

Results and Discussion1) Provide the complete spelling of BE4-gam.

Implemented.

2) Figure 1 The authors demonstrate highly efficient base editing with BE4-gam in 4 zebrafish cancer genes, with introduction of predictable, known cancer variants into each. It would be helpful to include the sequence of the entire target site with PAM for each gene in Figure 1.

Following reviewer 1 and 2 suggestions we have now added the edited base position in Figure 1. We have also reported the full sequence of the entire target site for each gene targeted in this paper in a new supplementary figure 1 with the annotation of the PAM and the edited C base (Figure 1—figure supplement 1).

Figure 1B-C: The authors introduce the wnt pathway activating mutation TCA (Serine) to TTA (Leucine) into ctnnb1 B-catenin using cytidine base editor BE4-gam. To demonstrate activation of wnt signaling ctnnb1 base editing was done in the wnt pathway reporter line Tg(tcf:GFP), and the results indicate an increase in GFP signal in the developing retina at 1dpf. Figure 1B upper panel: The low contrast makes it difficult to see the images – they appear to be low magnification of the head/anterior region of the embryo.

Yes, this is correct as the upper panel indeed shows a low magnification of the head region. Now to hopefully make this more evident, we delineated the outer edge of the embryos and added the orientation in the Figure 1 legend.

Quantification of the amount of GFP positive cells in control vs. ctnnb1 edited embryos or retina should be included, with the appropriate number of biological replicates noted.

Given the mosaic activation of Wnt signaling when using our approach, the rigorous quantification of GFP+ clones is only possible in the retina where the Tg(tcf:gfp) Wnt reporter line is not active at that stage. Overall in 3 independent experiments, we scored 39 embryos with increased GFP in the CNS and imaged 4 representative embryos at the confocal microscope in the retina for each condition. We observed an average of 12 GFP positive clones per retina while this was never detected in control injected Tg(tcf:gfp) embryos. These results are implemented in the main text and in the Figure 1B legend.

Figure 1C shows EditR analysis of the frequency of base editing; no details on the number of embryos analyzed was included in the results although the information is in the methods (10 embryos analyzed). Since this is a techniques paper, it would be helpful to include details on how base editing was analyzed: the number of embryos, if they were analyzed singly or pooled, how the PCR amplicons were sequenced to generate sequence data for EditR analysis. The frequencies reported in Table 1 are a little misleading, since as the authors note EditR drops out samples below 5% editing efficiency. For example, 5/8 = 63%, but in reality 5/10 = 50%.

We apologize for the confusion. The frequency in Table 1 referred to the number of single embryos analyzed as reported in Table 2 of randomly selected embryos. We have stated that in the Materials and methods section. Therefore, when we report 5/8 means that we analyzed only 8 embryos in which we could detected mutation with more that 5% of efficiency. We also added this number as they were presented in the Table 1 and Table 2 throughout the main text as requested.

Figure 1D-G The authors introduced nonsense C-T* mutations into three tumor suppressor genes with up to 86% efficiency. The sentence beginning "This last result is …" should be removed and replaced with an acknowledgement the similar efficiency reported by Zhao et al., 2020.

We adjusted our claims to include results from previous studies and discussed the different efficiencies obtained in these and our studies.

Amplicons were cloned and sequenced, and the data show either precise C-T conversion or wildtype sequence without indels. But the data in the panels D-G indicates other changes at some frequency; some clarification of this difference would be helpful.

We apologize if this was not clear, the amplicons were not cloned but directly sequenced. We made modifications in the figure legend specifying these low frequencies base pairs changes are below detection sensitivity of EditR and due to the background signal of the Sanger sequencing.

3) Figure 2. The BE4-gam base editor wasn't effective at introducing the oncogenic E62K mutation into Kras, or a Q8* nonsense mutation into the dmd tumor suppressor, so the authors tested the newer ancBE4max. It wasn't stated how it didn't work with BE4-gam, if it was a complete failure, or just indels were induced. Using the same sgRNAs ancBE4max was effective at introducing the edits at high efficiency. This is useful knowledge, if one base editor isn't working, the other could.

We only obtained non-edited sequences using the BE4-gam without INDELs for the Kras and Dmd genes. We clarified this point in the manuscript.

One of the most impactful and novel aspects of this study is the modification of ancBE4max to recognize a different PAM. The authors switched out the SpCas9 PIM domain (PAM-interacting motif) with Spymac PIM, which recognizes the PAM NAA. The data show relatively low efficiency of base editing at nras and tp53 (19% and 16%). Future work could optimize this base editor to increase efficiency.

This is indeed a very interesting point that will be the focus of future investigations.

4) Figure 3. Here the authors demonstrate efficient BE4-gam base editing in both somatic and germline targeting of the W577* mutation in the E3 ubiquitin ligase cbl. 4/15 adults carried the Cbl W577* in germ cells and transmitted it to 28% of their F1 offspring (44/153). Remarkably, all of the F1 offspring that inherited the precise W577* allele. Data from multiple loci is needed to obtain a frequency of germline transmission of precise base edited alleles.

We agree that our claim about transmission rate was wrong and corrected our result description in the main text. We now clearly indicate that this result reefers to a single F1 fish and we agree that more data would be needed to make a general statement.

24% of MZ cbl ^-/-^ adults show a short stature phenotype in comparison to MZ clb ^+/+^ siblings, and the authors claim this is a new model of dwarfism. Without details on the genetics used to establish the cbl mutant line, it's unclear if the phenotype is linked to the base edited allele or is in the background. If the fish were inbred without first establishing outcrossed F2 generation adults, it's possible this is due to background mutations. More rigorous genetic analysis or inclusion of details would strengthen the validity of the disease model.

The possibility that the dwarf phenotype comes from a background mutation was addressed through the generation of control fish that we obtained by keeping the wild-type siblings from the *cbl^+/-^* incross. The *cbl^+/+^* controls in Figure 3 are from the cross of these wild-type siblings, following the same procedure as for the *MZcbl^-/-^.* We never observed dwarf phenotypes in these *cbl^+/+^* fish neither in the *cbl^+/-^* and *cbl^-/-^* fish. The dwarf phenotype was also only observed in *MZ cbl^-/-^* embryos derived from two independent crosses. It should be noted that we have now kept this *cbl^+/-^* line for 5 generations in our fishroom in the same genetic background and the dwarf phenotype was only observed when *MZ cbl^-/-^* embryos were raised. In addition, the dwarf phenotype was neither observed in our wildtype stocks from which the *cbl ^-/+^* line was derived. If a background mutation happened to be linked to our mutant allele, since the dwarf phenotype was only observed in *MZ cbl^-/-^* and not in zygotic *cbl^-/-^,* the background mutation would be in a second independent maternal zygotic mutant gene which we cannot formally exclude but seems unlikely. We now discussed these data in our manuscript as suggested.

Reviewer #2 (Recommendations for the authors):My most significant concern has to do with the way editing data is described throughout the manuscript. To take a specific example, one might infer from Figure 1C (including figure legend) that substitution of Serine 33 for Leucine in b-catenin is >70% efficient. But is it? Based on Supplementary Table 3, editing of C13 was only detected in 5/8 embryos analyzed (62.5%), with efficiencies of 73%, 40%, 25%, 16% and 11%. Thus, average editing efficiency in positive embryos is 33%. If one were to include negative embryos, editing efficiency would be ~19% – which is still very good but quite different from 70%. I believe editing efficiency ranges must be presented in the main figures, perhaps as dot charts.

We agree with the comment and for this reason we presented the efficiency we obtained in single embryos in the Table 2. In the text we are careful to indicate the efficiency is in the best case obtained and we state always state the number of embryos in which we observed editing at detectable levels.

Along the same lines, it is not clear how the embryo imaged in Figure 1B was selected. How many embryos were scored for ectopic activation of wnt signaling reporter and how many were found positive? To what extent (for example, what were the numbers of GFP-positive RPCs in each)? Does the photographed embryo represent "best case" or "average"?

Given the mosaic activation of Wnt signaling when using our approach, the rigorous quantification of GFP+ clones is only possible in the retina where the Tg(tcf:gfp) Wnt reporter line is not active at that stage. Overall in 3 independent experiments, we scored 39 embryos with increased GFP in the CNS and imaged 4 representative embryos at the confocal microscope in the retina for each condition. We observed an average of 12 GFP positive clones per retina while this was never detected in control injected Tg(tcf:gfp) embryos. These results are implemented in the main text and in the Figure 1B legend.

Given that C to T base editors may mutate any Cs within the editing window, it is important to present data from Sanger sequencing of individual amplicons, similarly to data presented for tek Q94*, bap1 Q273*, tp53 Q21*, tp53 Q170* (Figure 1D-G) and cbl W577* (Figure 3B). What about the ctnnb1 S33L amplicon from Figure 1C? Do the desired change (73% efficiency) and the silent bystander substitution (74% efficiency) co-occur at random? If that were the case, one would expect ~50% of sequences with both changes and ~20% of sequences with either single change. Or do both changes co-occur at much higher than random probability, as Figure 3C suggests? This would be extremely important if the second change was not silent but introduced a missense or nonsense mutation. For those scenarios, perhaps embryo 2 and embryo 5 (Supplementary Table 3) are worth highlighting as C15 substitution is not observed in them? This applies to other edits where significant fraction of additional C->T substitutions is observed: kras (Figure 2A) and rb1 (Figure 2D).

We agree these are important comments for which we reported all the raw data in Table 2. In general, in our hands the frequency of mutations in other C bases present in the editing window depends on their localization, with a highest efficacy for the base that are at the center of this window. As the reviewer correctly pointed out, in embryos 2 and 5, we observe only editing on the wanted C at position 13 with no editing of the C at position 15, suggesting that these are independent events.

Reviewer #3 (Recommendations for the authors):I have only relatively minor suggestions to improve the clarity of presented data.Figure 1B: The number of embryos with increased GFP/ total examined should be indicated in the legend.

Given the mosaic activation of Wnt signaling when using our approach, the rigorous quantification of GFP+ clones is only possible in the retina where the Tg(tcf:gfp) Wnt reporter line is not active at that stage. Overall in 3 independent experiments, we scored 39 embryos with increased GFP in the CNS and imaged 4 representative embryos at the confocal microscope in the retina for each condition. We observed an average of 12 GFP positive clones per retina while this was never detected in control injected Tg(tcf:gfp) embryos. These results are implemented in the main text and in the Figure 1B legend.

Figure 1C-G. It took this reviewer a long time to understand how these figures worked. Once I did, I felt it was a clear way of showing the data. However, some additional description in the legend would have been very helpful for orientation. The ACTG in the top corner was initially mistaken for sequence, rather than a color code, which likely contributed significantly to my early confusion. It should also be made clearer that the numbers represent % at that site, and the box color coding system should be clearly stated.

To facilitate the understanding of our figures, we changed the ACTG legend for all the chromatograms and specified in the legends the meaning of the color codes, hoping that it is now less confusing.

Figure C-G. It is not clear exactly what this data is from. Is it one embryo, a pool of embryos? How many sequences analyzed? A general strategy for sequencing is discussed in the Materials and methods, but it would be helpful to know the specifics of the data presented in this figure for each mutant.

We agree with the comment and for this reason we presented the efficiency we obtained in single embryos in the Table 2. In the text we are careful to indicate the efficiency we report is in the best case obtained and we state always the number of embryos in which we observed editing at detectable levels. The chromatogram always refers to the efficiency reported for the first embryos provided in the first column of the Table 2 which correspond to the efficiency reported in Table 1. We added these details in the legend of the figures.

It is not clear whether the python script used for sgRNA design will be made available.

The python script used for sgRNA design is made as an additional file.

In figure D, I am unclear how one can have wildtype +/+ siblings in a cross with maternal zygotic -/- mutants. Wouldn't the mother have to have been -/-?

We apologize for the mistake, it is indeed not correct. These are not siblings but wildtype controls derived from incross of the wildtype siblings of the zygotic *cbl^-/-^* fish. We corrected this mistakes in the manuscript.